# Host-directed broad-spectrum immunotherapeutic strategy for respiratory infections: Heat-killed *Caulobacter crescentus* (HKCC) as an innate-immune based biotherapeutic/postbiotic

**Shanika Werellagama[1], Raj S. Patel[1], Nancy Gupta[2¤a], Satish Vedi[2¤a], Diana Duque[3], Jegarubee Bavananthasivam[3], Rakesh Kumar[2¤b¤c], Anh Tran[3,4], Babita Agrawal**[1]*

**1** Department of Surgery, Faculty of Medicine and Dentistry, College of Health Sciences, University of Alberta, Edmonton, Alberta, Canada, **2** Department of Laboratory Medicine and Pathology, Faculty of Medicine and Dentistry, College of Health Sciences, University of Alberta, Edmonton, Alberta, Canada, **3** Infectious Diseases, Human Health Therapeutics Research Centre, National Research Council Canada, Ottawa, Ontario, Canada, **4** Department of Biochemistry, Microbiology and Immunology, University of Ottawa, Ottawa, Ontario, Canada

¤a Current Affiliations: Independent Scientist, Edmonton, Alberta, Canada,
¤b Current Affiliations: Professor (Retd.), University of Alberta, Edmonton, Alberta, Canada,
¤c Current Affiliations: ImMed Biotechnologies, Edmonton, Alberta, Canada
* bagrawal@ualberta.ca

## Abstract

Respiratory infections are among the leading causes of illness and death. The lack or limited effectiveness of vaccines for many respiratory pathogens underscores the urgent need for alternative, broadly protective strategies. While adaptive immunity is commonly used in vaccine development, the therapeutic potential of activating and enhancing innate immunity remains underutilized. Innate immune–focused interventions could offer rapid, pathogen-independent protection, bridging the gap until pathogen-specific responses develop. Therefore, exploring new broad-spectrum innate immune-targeting immunomodulatory agents can be an effective way to prevent and treat respiratory infections. Heat-killed *Caulobacter crescentus* (HKCC) is a potential innate immune modulator. We tested its preventive and therapeutic effects against key respiratory bacterial (*Mycobacterium avium*, *Mav*) and viral (SARS-CoV-2 and influenza) infections in animal models. Notably, intranasal or oral treatment with HKCC significantly lowered bacterial loads in mice infected with *Mav*, while activating mucosal innate immune responses and enhancing downstream cellular and antibody responses. Additionally, decreased viral loads and improved pathogenesis were seen in mice and hamsters infected with influenza and SARS-CoV-2, respectively. HKCC showed promising ability to induce strong, localized, and systemic immunity, making it a compelling candidate for developing as a human preventive or adjunct therapy for multiple respiratory diseases.

**Data availability statement:** All relevant data are in the manuscript and its supporting information files.

**Funding:** This work was supported by Canadian Institutes of Health Research grants PJT165854, PS173314 and PJT198006 to BA. The funders had no role in the design of the study; in the collection, analyses, or interpretation of data; in the preparation of the manuscript, or in the decision to publish the results.

**Competing interests:** I have read the journal's policy and the authors of this manuscript have the following competing interests: BA and RK are coinventors on several patents issued worldwide on HKCC adjuvant, and are cofounders of ImMed Biotechnologies.

## Author summary

Respiratory infectious diseases, including major threats like mycobacteria, influenza, and SARS-CoV-2, remain a significant global public health concern. While our body's initial defense system at the site of entry of these pathogens—the mucosal innate immune response—is key to quickly clearing or lessening early infections, we currently lack effective and safe therapeutic candidates that modulate these innate responses. Consequently, new and comprehensive preventive or therapeutic tools are necessary to control these respiratory infections. Our research has shown that inactivated, non-pathogenic *Caulobacter crescentus*, delivered through nose or mouth (mucosally), functions effectively as an innate immune modulator. In preclinical models, it effectively boosts key immune players (including innate lymphoid cells and neutrophils), enhances their function, and stimulates robust antibody production. Crucially, this immune activation led to promising reductions in both bacterial and viral loads and resulted in significantly milder disease progression. In summary, prompt use of heat-killed *C. crescentus* offers an effective, ready-to-use strategy to enhance the body's primary defenses against a broad range of bacterial and viral respiratory pathogens. It is a highly viable candidate for translation into future human clinical trials.

## Introduction

Respiratory infectious diseases are a leading cause of mortality worldwide. According to the World Health Organization (WHO), tuberculosis (TB), lower respiratory infections such as influenza A viruses, and COVID-19 rank among the top ten causes of death globally [1,2].

Tuberculosis is an age-old disease caused by *Mycobacterium tuberculosis* (*Mtb*) and *M. avium* complex (MAC) that continues to result in a significant number of deaths worldwide, especially in endemic regions. The WHO reported that in 2023, 10.8 million people were ill with TB, and 1.25 million deaths occurred due to TB [3]. Drug resistance to available TB medications is already a global crisis [4], and relying on current drugs for treatment is not a sustainable long-term solution. One quarter of the world's population is infected with TB bacteria, and some of these individuals have a higher risk of developing active TB disease [5]. Prevention is crucial to halt TB transmission. TB preventive treatment is considered one of the most important public health measures to protect individuals and communities from the disease. Bacille Calmette-Guérin (BCG), the only vaccine available for TB since its development, celebrated its centenary. However, BCG provides only limited protection for adults, with vaccine efficacy ranging from 0% to 80%, which presents significant challenges to TB control [6]. The WHO has emphasized the need for an effective prophylactic and immunotherapeutic vaccine for adult TB [4]; however, efforts to develop preventive and/or therapeutic vaccines have achieved limited success and face major obstacles.

MAC is a group of mycobacteria that can cause lung infections as well as infect the lymph nodes, bones, joints, skin, and soft tissues, and may lead to disseminated disease. MAC infections, particularly *M. avium* (*Mav*), an environmental organism, can cause severe illness and even death in immunocompromised individuals and are associated with patients having Human immunodeficiency virus (HIV), cancer, bronchiectasis, lung cancer, cystic fibrosis (CF), and chronic obstructive pulmonary disease (COPD). *Mav* infections pose a significant challenge to the clinical management of TB in HIV patients and contribute to high mortality rates. Furthermore, *Mav* is intrinsically resistant to most first-line anti-TB drugs and quickly develops resistance [7].

Influenza A virus is a highly destructive seasonal pathogen that causes high mortality and illness worldwide each year. Influenza infections result in millions of cases, 3–5 million severe cases, and roughly 650,000 deaths annually [8]. In addition to seasonal infections, influenza A virus variants with pandemic potential emerge sporadically in humans. Antigen drift and shift variants of the influenza A virus continue to challenge vaccine development, as the influenza vaccine has shown effectiveness from 19% to 60% during 2009–2025 [9]. Additionally, the current influenza vaccine shows different levels of effectiveness across age groups [10]. The limitations of the current influenza vaccines have resulted in unexpected hospitalizations and deaths worldwide.

The SARS-CoV-2 infections causing the COVID-19 pandemic began at the end of 2019. Although the pandemic has subsided, infections with SARS-CoV-2 and its emerging variants continue to circulate in the population. By mid-2025, approximately 7 million deaths have been attributed to SARS-CoV-2 infections [11]. The global fear of the pandemic drove vaccine development at an unprecedented speed during 2020–21. The authorized vaccines against SARS-CoV-2 help reduce the severity of the disease and decrease mortality [12]; however, reduced vaccine efficacy against emerging variants has necessitated the development and implementation of updated periodic vaccine boosters.

The primary approach to preventing pandemic-prone viral infections relies on herd immunity achieved through widespread vaccination efforts. However, antiviral drugs can serve as a viable option in managing the disease. Despite their necessity, effectiveness and resistance issues are major concerns for most antiviral drugs [13]. For instance, the limited and uncertain effectiveness of some major antiviral drugs against COVID-19 and influenza has been reported previously [14,15].

The development of vaccines against respiratory pathogens, which offer long-lasting protection against emerging mutated strains and provide mucosal immunity, remains a significant ongoing challenge. Choosing immunogenic antigens and a safe adjuvant system is an important yet complex aspect of vaccine development against these pathogens [16]. Next-generation vaccines face additional difficulties due to factors like antigen selection, vaccination route, booster frequency, and host diversity [17]. Beyond scientific challenges, vaccine hesitancy and misinformation also pose serious obstacles to the implementation of new vaccines.

Naturally protective immune responses, involving both innate and adaptive immunity, occur in distinct phases. The innate immune response is crucial for combating respiratory pathogens during the early stages of infection. Notably, only 10% of immunocompetent individuals exposed to mycobacteria may develop active infections, indicating that 90% have effective immune mechanisms, especially innate immunity, to eliminate the infection early on [18]. Epidemiological studies also reveal a strong link between age and infections with influenza and SARS-CoV-2 that are difficult to resolve or remain unresolved [19]. Younger children with robust innate immunity tend to resist developing symptomatic disease and death from viruses [20]. This age-related difference in respiratory virus infections is primarily attributed to a decline in innate immunity among middle-aged and older adults [21]. Generally, the respiratory tract epithelium, macrophages, dendritic cells (DCs), natural killer (NK and NKT) cells, and innate lymphoid cells (ILCs) form the dynamic innate immune component of the respiratory mucosal immune system [18]. These cells play a key role in defending against infection, maintaining immune homeostasis, and preventing progression to clinical symptoms during respiratory infections and lung pathology. Therefore, well-coordinated immune responses, including innate immunity, can largely eradicate or control respiratory infections, leading to milder disease manifestations [22].

Emerging evidence also indicates that various commensal microbes living in the respiratory tract and lungs are vital for immune health and contribute to maintaining the proper immune function and resistance to infectious diseases [23]. Beneficial bacterial species, especially in their non-replicative, killed, or postbiotic forms, may play a key role in strengthening innate immunity, maintaining homeostasis, and protecting against respiratory infections [24]. Therefore, developing new host-targeted immunotherapies that enhance innate immunity, particularly in the respiratory mucosa, is an ideal approach that can be used both as a preventive (pre-exposure) and a therapeutic (post-exposure) to control deadly respiratory infections and limit their spread. Our earlier studies have shown that the heat-killed form of *Caulobacter crescentus* (HKCC), a freshwater, non-pathogenic bacterium, acts as an innate immunomodulatory agent [25]. It has also shown strong potential as a novel mucosal vaccine adjuvant for SARS-CoV-2 [26].

In this study, we aimed to investigate HKCC as a standalone host innate immune-based immunotherapy for the prevention and/or treatment of chronic and acute respiratory bacterial and viral infections, *Mav*, SARS-CoV-2, and influenza virus. Strikingly, the results showed that across all three infection models, HKCC produced a strong therapeutic effect by reducing infectious load and easing disease severity. Notably, HKCC triggered balanced innate immune responses, positioning it as an effective mucosal immunomodulatory agent. These studies highlight the promise of HKCC as a broad immunomodulatory agent against major respiratory pathogens and could serve as a viable option for their prevention and treatment in humans.

## Results

### Efficacy of HKCC against *Mycobacterium avium* infection in mice

To determine the effect of different schedules and routes of administration, we treated BALB/c mice with HKCC intranasally or orally according to the timeline shown (Fig 1A). Then, the mice were challenged intravenously with $5 \times 10^5$ CFU/50 µL of logarithmically growing *Mav*, and bacterial loads were measured in three organs: lungs, liver, and spleen. Compared to the untreated-infected control group, treatment with HKCC significantly reduced the *Mav* loads in all three organs and across all three treatment groups examined. Among the two different schedules of intranasal HKCC treatment, 8 and 1 days before challenge showed a significantly lower bacterial load compared to 21 and 8 days. When comparing oral versus intranasal routes, intranasal administration resulted in a noticeably greater reduction in *Mav* load in all organs (Fig 1B).

To assess the impact of HKCC treatment on *Mav* challenge–induced lung pathologies and disease severity, parts of the collected lungs were fixed, sectioned, and prepared for H&E staining to perform histopathological analysis. *Mav*-infected untreated mice showed severe inflammatory lesions, reduced free alveolar air space, unorganized or early-stage granuloma lesions, epithelial cell hyperplasia, tissue necrosis, cell debris, and infiltration of red blood cells and immune cells (Fig 1C). In contrast, all three treatment groups exhibited milder pathologies compared to the untreated-infected control group (Fig 1C). To evaluate overall lung tissue severity, we used a specific scoring system that included alveolar space unavailability, alveolar epithelial hyperplasia, and infiltration of red blood cells and immune cells (severity scores ranged from 0 to 5). Different stages of granulomas were also identified and counted. The scores and granuloma counts were summed to calculate the lung severity score for each group. Importantly, all HKCC-treated groups had significantly lower lung severity scores compared to the untreated group (Fig 1D). Intranasal administration of HKCC on days 8 and 1 before the challenge resulted in the lowest lung pathology severity, with very little alveolar epithelial hyperplasia, less granuloma development, and preserved air space (Fig 1C).

### Analysis of innate lymphoid cells and their functional profiles in HKCC-treated *Mav*-infected mice

Based on the promising antimicrobial effects of HKCC treatment in *Mav*-infected mice (Fig 1), we examined whether HKCC treatment can trigger protective innate immune responses using flow cytometry. HKCC treatment via the intranasal route increased the number of ILCs (CD3⁻CD127⁺) and NK cells (CD3⁻CD127⁺CD49⁺) in the lungs compared to the untreated-infected group (Fig 2A I and 2A II). When comparing the intranasal and oral routes of HKCC administration, oral

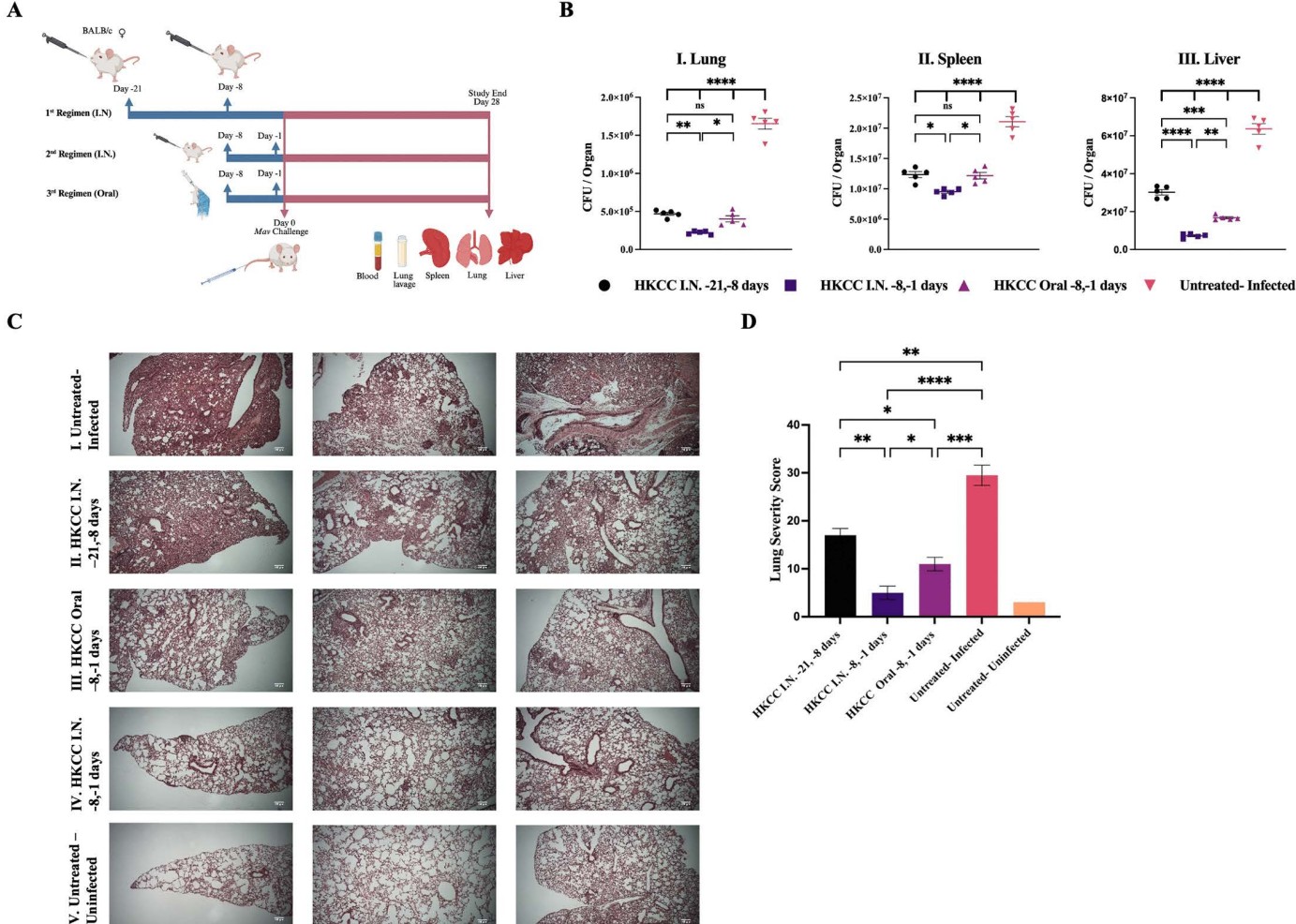

**Fig 1. Efficacy of HKCC against *Mav* infection in mice. (A)** Schematic timeline of various HKCC treatment regimens and sample collection (Created with Biorender, https://BioRender.com/pia4ous). **(B)** Six- to eight-week-old female BALB/c mice (n = 5 per group) were treated with 50 × 10⁶ CFU of HKCC intranasally (total volume: 30 μL, 15 μL in each nostril) or orally (50 × 10⁶ CFU in 200 μL) before being infected with *Mycobacterium avium* (*Mav*) (5 × 10⁵ CFU in 50 μL) intravenously. Four weeks after the challenge, mice were euthanized, and various samples (blood, lung lavage, lung, spleen, liver) were collected. Bacterial loads were measured in **(I)** lungs, **(II)** spleen, and **(III)** liver using CFU assays. **(C)** Representative hematoxylin and eosin **(H&E)**-stained lung tissue images and the scale bars represent 100μM, and **(D)** scoring of H&E-stained lung sections in a double-blind manner based on three criteria described in the methods section, along with the number of granuloma lesions. Results are expressed as the mean ± standard error of the mean (SEM) of CFU counts from individual mice. Statistical significance (*$p < 0.05$; **$p < 0.01$; ***$p < 0.001$) was determined using one-way ANOVA followed by Tukey's test.

delivery did not affect ILC and NK cell counts and resulted in even fewer cells than the untreated-infected group (Fig 2A II). In the lungs, intranasal HKCC treatment given 8 and 1 day before challenge also heightened the expression of RORγt, T-bet, IFNγ, IL-17A, and IL-13, as shown by the increased color intensity, compared to the untreated-infected control (Fig 2A III). Moreover, histograms comparing transcription factors and cytokines among treatment groups revealed a distinct peak shift in RORγt, IL-17A, and IL-13 in the group that received intranasal HKCC 8 and 1 day before challenge (Fig 2A IV). These findings suggest that intranasal HKCC treatment given 8 and 1 day prior to infection enhances type 2 and 3 ILC responses (Fig 2A III and 2A IV). The group that received HKCC intranasally 21 and 8 days before challenge elicited high level of IFNγ, and slightly reduced IL-17A and IL-13 expression. In contrast, the untreated-infected control showed

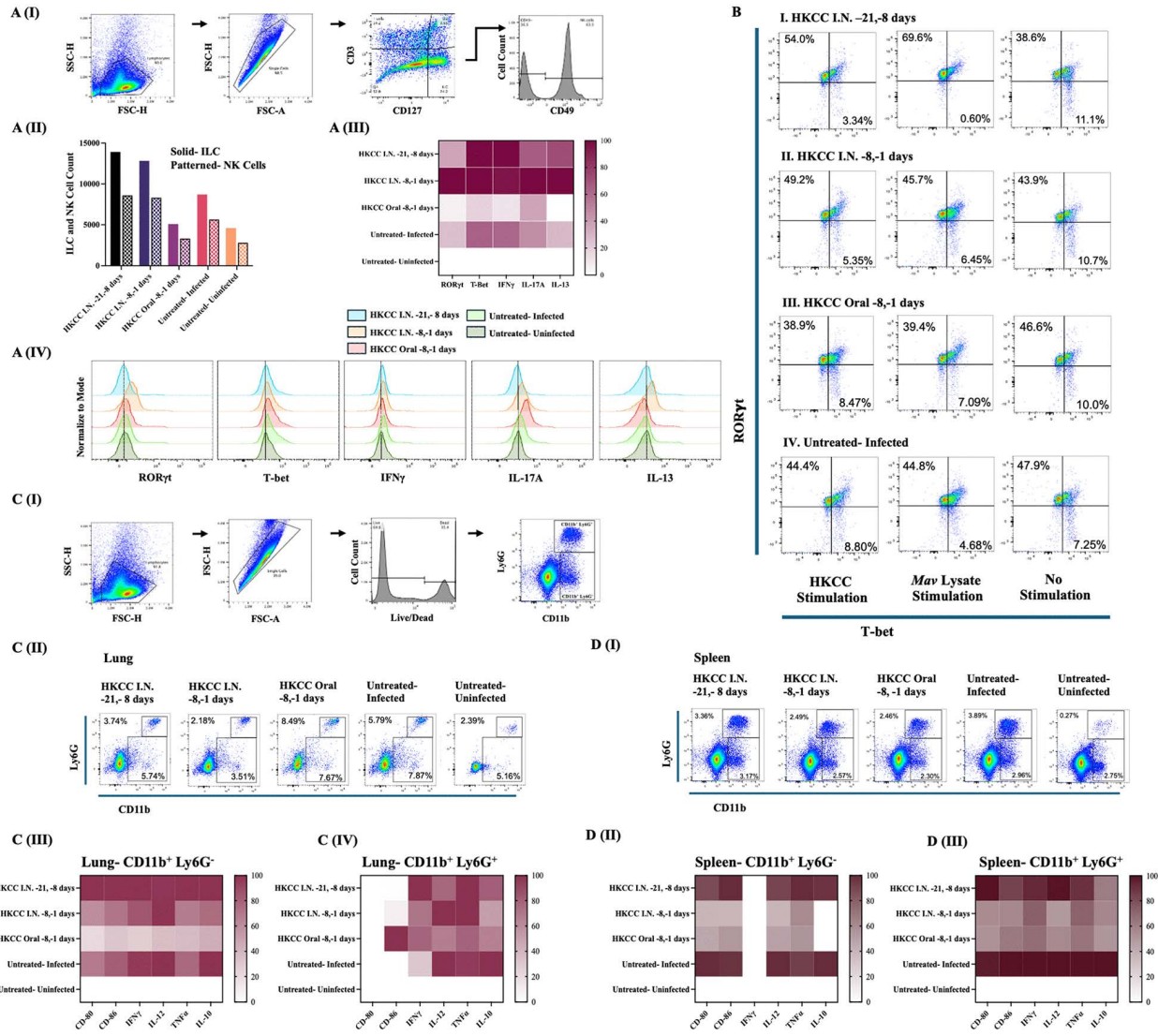

**Fig 2. Flow cytometry analysis of lung and spleen cells in HKCC-treated *Mav*-infected mice. (A)** Lung lymphocytes and splenocytes were seeded in a 24-well plate and stimulated with PMA (50 ng/mL) and ionomycin (500 ng/mL) overnight at 37°C. Then, brefeldin A (1.5 μg/mL) was added, and cells were incubated for an additional 4 hours. Subsequently, staining was performed as required for each analysis. **(I)** Flow cytometry gating strategy for ILCs and NK cells. **(II)** Number of ILCs and NK cells in different groups. **(III)** Heat map of ILCs across groups showing various transcription factors and cytokines, with expression levels normalized to those in the untreated-uninfected control group. **(IV)** Histogram comparing transcription factors and cytokine expression across different groups. **(B)** Splenocytes ($2 \times 10^6$ Cells/well) from each treatment group were stimulated with HKCC ($50 \times 10^6$ CFU/well) or *Mav* lysate ($1 \times 10^6$ CFU/well) *in vitro*, and transcription factors were measured using flow cytometry. Changes in expression of RORγt and T-bet are shown under medium control (right), *Mav* lysate (middle), and HKCC (left) stimulation. **(C) (I)** Gating strategy for CD11b⁺ and CD11b⁺Ly6G⁺ cells. **(II)** Proportion of CD11b⁺ and CD11b⁺Ly6G⁺ cells in the lungs across groups. **(III)** Heat map of costimulatory molecules and different cytokine profiles of CD11b⁺ cells in the lungs. **(IV)** Heat map of costimulatory molecules and cytokine profiles of CD11b⁺Ly6G⁺ cells in the lungs. **(D) (I)** Proportion of CD11b⁺ and CD11b⁺Ly6G⁺ cells in the spleens across groups. **(II)** Heat map of costimulatory molecules and cytokine profiles of CD11b⁺ cells in the spleens. **(III)** Heat map of costimulatory molecules and cytokine profiles of CD11b⁺Ly6G⁺ cells in the spleens.

reduced cytokine expression compared to the two intranasal treatment groups (Figs 2A and 1C). Conversely, oral administration demonstrated a reduced expression profile of ILCs compared to the untreated-infected group (Fig 2A III), consistent with the lower ILC numbers.

To further evaluate the lineage commitment and/or training of systemic ILC subsets in treated mice, we stimulated splenocytes from treated mice for 4 days with either HKCC lysate or *Mav* lysate separately. For controls, an equal number of cells were cultured without stimulation in each treatment group. Unlike the untreated-infected and orally HKCC-treated groups, the intranasal treatment groups in both schedules showed induction of RORγt expression upon restimulation with HKCC and *Mav* lysate, demonstrating a trained feature dependent on microbial stimulation but not antigen specificity (Fig 2B I and 2B II). The group that received HKCC intranasally on days 8 and 1 before the challenge had 43.9% and 10.7% of RORγt- and T-bet-expressing cells, respectively, under unstimulated conditions. When those splenocytes were stimulated with HKCC or *Mav* lysate, the percentages increased to 49.2% and 45.7% of RORγt-expressing cells and decreased to 5.35% and 6.45% of T-bet-expressing cells, respectively (Fig 2B II). Interestingly, the orally HKCC-delivered group and the untreated-infected groups exhibited a similar pattern of ILC differentiation (Fig 2B III and 2B IV). Both groups showed reduced RORγt transcription factor expression and either decreased or maintained T-bet expression upon stimulation with HKCC or *Mav* lysate. Untreated-infected splenocytes displayed 47.9%, 44.8%, and 44.4% RORγt expression in the absence of stimulation, with *Mav* lysate, and with HKCC lysate, respectively. Additionally, 7.25% of cells were positive for T-bet without stimulation. Upon stimulation with *Mav* lysate and HKCC lysate, T-bet expression was 4.68% and 8.80%, respectively (Fig 2B IV).

### Evaluation of myeloid cell transmigration and cytokine expression in HKCC-treated *Mav*-infected mice

Granulocytes, cells of myeloid origin, play a key role in innate immunity against infections [27]. CD11b+ granulocytes, especially those positive for Ly6G, infiltrate infection sites and create a supportive environment for pathogen growth [28]. Additionally, they have a distinct ability to communicate with T cells and modulate their responses [27]. Therefore, we examined the CD11b+ and CD11b+Ly6G+ subpopulations across the treatment groups using flow cytometry in both lungs and spleen (Fig 2C I). In the lungs, a higher percentage of CD11b+ and CD11b+Ly6G+ infiltration was observed in the untreated-infected (7.87% and 5.79%), compared to the groups treated with HKCC intranasally on both schedules (days -21, -8, and -8, -1) (Fig 2C II). The group that received HKCC intranasally 8 and 1 day before the challenge, which showed the least bacterial load and lung pathology, also had the lowest number of CD11b+ and CD11b+Ly6G+ cells (3.51% and 2.81%, respectively) (Fig 2C II).

Next, each subset (CD11b+ and CD11b+Ly6G+) was further analyzed to assess their cytokine expression. Results were normalized to the untreated-uninfected control and presented as a heatmap. The CD11b+ population showed that, in the intranasal 21- and 8-day schedules, the costimulatory molecules CD80/86 were expressed at relatively high levels, along with high expression of other effector cytokines (IFNγ, IL-12, and TNFα) (Fig 2C III). However, this group was associated with greater IL-10 expression. The untreated-infected group displayed a similar pattern, co-expressing effector cytokines such as IFNγ and IL-12, along with high IL-10 levels (Fig 2C III). Interestingly, intranasal and oral treatments with HKCC 8 and 1 days before the challenge showed a distinct pattern, characterized by decreased IL-10 levels and moderate levels of pro-inflammatory cytokines such as IFNγ, IL-12, and TNFα (Fig 2C III). Analysis of the CD11b+Ly6G+ subset revealed that the untreated-infected control had an immunosuppressed profile with the highest IL-10 intensity and the lowest IFNγ intensity in the heatmap (Fig 2C IV). Conversely, all three HKCC treatment regimens reduced IL-10 expression and increased IFNγ levels (Fig 2C IV). Oral HKCC treatment showed the highest CD86 expression on CD11b+Ly6G+ cells; the reasons and implications of this observation remain unclear. Notably, intranasal treatment with HKCC 21 and 8 days before challenge showed higher levels of IFNγ and TNFα, while treatment 8 and 1 days prior to challenge led to elevated IL-12 and TNFα, with lower IL-10 levels (Fig 2C IV). Additionally, oral HKCC treatment exhibited higher IFNγ levels and lower IL-10 levels (Fig 2C IV). Overall, groups with elevated IFNγ, IL-12, TNFα, and IL-10 seem to support *Mav* growth and promote inflammatory lung pathology, as seen in the untreated-infected mice (Figs 1C and 2C).

In the spleen, dot plots showed that both intranasal and oral treatments with HKCC on days -1 and -8 resulted in the lowest percentage of myeloid cell infiltration, at 2.30-2.57% and 2.46-2.49% for CD11b+ and CD11b+Ly6G+ cells,

respectively (Fig 2D I). In contrast, the highest percentages of CD11b$^+$ and CD11b$^+$Ly6G$^+$ cells were observed in the group treated intranasally with HKCC on days -21 and -8 (3.17% and 3.36%, respectively), compared to the untreated-infected group (2.96% and 3.89%, respectively) (Fig 2D I). The heatmap of the CD11b$^+$ population in both treatment groups (intra-nasal and oral on days -8 and -1), compared to the untreated-uninfected group (Fig 2D II), showed a similar pattern to that observed in the lungs. Although costimulatory molecules and effector cytokines are elevated in the group treated intrana-sally with HKCC on days 21 and 8 before challenge, this group exhibited reduced IL-10 expression in CD11b$^+$Ly6G$^+$ cells in the spleen (Fig 2D III). When comparing groups treated with HKCC intranasally and orally at days -8 and -1, higher expressions of CD80, CD86, IL-12, and TNFα were observed in the oral treatment (Fig 2D II). The spleen CD11b$^+$Ly6G$^+$ subset in the untreated-infected group showed an equally high expression of various costimulatory molecules, effector cytokines, and IL-10, as also seen in the lungs (Fig 2C IV). Overall, HKCC administration substantially reduced IL-10 expression, along with varying levels of other cytokines (Fig 2D III).

### Induction of systemic and mucosal antibody titers and correlation between antibody titers and mycobacterial burden in HKCC-treated *Mav*-infected mice

Stimulation of innate immunity upon exposure to an infection can lead to the downstream induction of protective antibody responses. Therefore, to determine whether the HKCC treatment elicits systemic and mucosal antibody responses against mycobacteria, we measured the systemic IgG and mucosal IgA antibody titers in serum and lung wash samples, respec-tively, against lysates of *Mav* and the related species *M. bovis* (BCG). Intranasal administration of HKCC on days 8 and 1, just before challenge, significantly increased *Mav*- and BCG-specific IgG titers compared to the untreated-infected control group (Fig 3A I and 3A II). Similarly, oral administration of HKCC enhanced systemic IgG titers specific to *Mav* and BCG compared to the untreated-infected group. In contrast, the group treated with HKCC intranasally on days -21 and -8 showed much lower *Mav*-specific IgG antibody levels than those specific for BCG (Fig 3A I). Interestingly, very small to negligible quantities of IgG antibodies specific to HKCC were detected in all three treatment groups (Fig 3A III). These results suggest that mucosal administration of HKCC shortly before the *Mav* challenge prominently induces pathogen-specific antibodies.

In the lung lavages, we observed that *Mav*- and BCG-specific IgA antibody titers increased across all treatment groups compared to the untreated-infected group; however, this increase was statistically significant only in the group that received HKCC orally (Fig 3B I and 3B II). Notably, similar to IgG, HKCC-specific IgA was also negligible in lung lavages across all three treatment groups (Fig 3B III).

Next, we evaluated the potential role of IgG and IgA antibodies in reducing bacterial loads in different organs. Correla-tion analysis showed that IgG antibody titers specific for *Mav* and BCG negatively correlated with bacterial load in the lungs (*Mav*: r = -0.5762, p = 0.0078; BCG: r = -0.4475, p = 0.0479) and spleens (*Mav*: r = -0.4626, p = 0.04; BCG: r = -0.4957, p = 0.0262), with these correlations being statistically significant (Fig 3C I and 3C II). In the liver, a significant inverse correlation was found only with *Mav*-specific IgG titers (r = -0.5684, p = 0.0089) (Fig 3C III). We also correlated mucosal IgA antibody titers with CFU counts, where negative correlation trends for *Mav* and BCG specific IgA titers were seen across all organs (lungs, spleen, and liver) (Fig 3D I-3D III). However, only the correlation between *Mav*-specific IgA titers and CFU in the lungs reached statistical significance (r = -0.4681, p = 0.0374) (Fig 3D I). Overall, these results support the idea that innate immune stimulator HKCC induces production of pathogen-specific systemic and mucosal antibodies, which may help control the infection during its early stages.

### Efficacy of HKCC against the influenza (H1N1) infection in mice

To evaluate the effect of the innate immune stimulator HKCC against influenza infections, female BALB/c mice were treated with HKCC ($50 \times 10^6$ CFU/mouse), administered either intranasally or orally, twenty-four hours before (Fig 4A) or after (Fig 4B) infection with the mouse-adapted PR8 (H1N1) influenza virus (60 PFU/mouse). The body weight of individual mice was tracked from the time of infection and for five days afterward, and viral loads were measured in lung homogenates. Mice

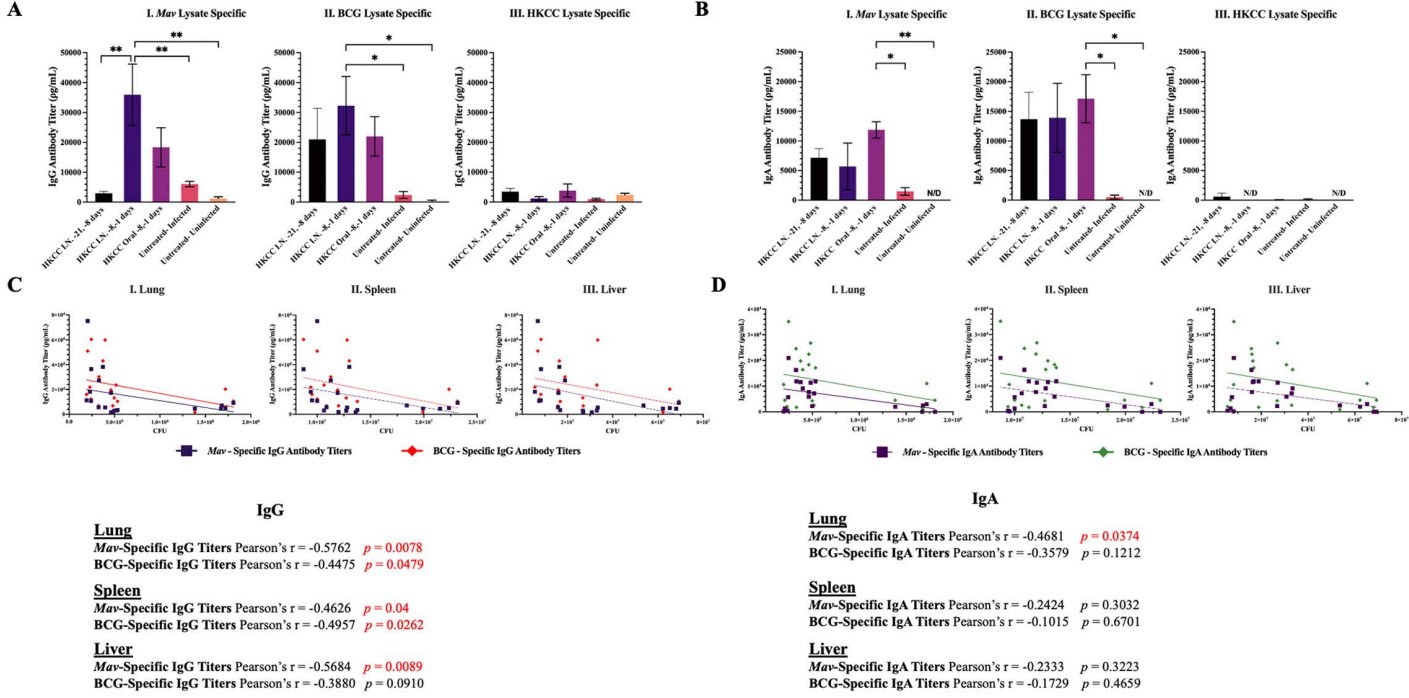

**Fig 3. Induction of systemic and mucosal antibody titers following HKCC treatment in *Mav*-infected mice.** After the different HKCC treatment protocols shown in Fig 1A, cardiac blood and lung lavage were collected from each mouse after euthanasia. Serum samples (1:100) and lung lavage samples (1:1) were diluted separately in sterile PBS, then used in antibody ELISA. (**A**) Systemic IgG antibody titers; (**B**) Mucosal IgA antibody titers specific for (**I**) *Mav*, (**II**) BCG, and (**III**) HKCC lysates. (**C**) Pearson's correlations between serum IgG titers and (**D**) mucosal IgA titers, as well as CFU counts in (**I**) lungs, (**II**) spleen, and (**III**) liver. Data are presented as mean ± SEM, with each data point representing an individual mouse (n = 5 per group). Results are shown as mean ± SEM of individual mice. Statistical significance (*$p < 0.05$; **$p < 0.01$; ***$p < 0.001$) was determined by one-way ANOVA followed by Tukey's test.

treated with HKCC either intranasally or orally 24 hours before infection showed a significant increase in body weight from days 1–5, compared to the untreated group, which lost 7% of their weight during the same period (Fig 4A II). Both intranasal and oral administration of HKCC resulted in a significant reduction in lung viral loads five days post-infection, compared to untreated mice (Fig 4A III). Intranasal treatment with HKCC, twenty-four hours before infection, caused more than a 2-log reduction in lung viral loads, while oral treatment with HKCC led to over a one-log reduction (Fig 4A III).

Interestingly, in the second set of experiments, where HKCC treatment was given twenty-four hours after infection, the body weights of all untreated and HKCC-treated groups declined over five days post-infection (Fig 4B II). However, an average weight loss of 8%, 12%, and 20% was observed for the HKCC (intranasally treated), HKCC (orally treated), and untreated groups, respectively (Fig 4B II). Notably, both HKCC administered intranasally and orally resulted in significantly less weight loss after infection compared to the untreated group (Fig 4B II). Interestingly, HKCC administration after viral infection via both intranasal and oral routes significantly reduced lung viral loads by more than half a log compared to the untreated group (Fig 4B III). Overall, a single intranasal or oral treatment with HKCC, in both pre- and post-viral challenge scenarios, proved effective against influenza infection by reducing body weight loss and viral loads in the lungs.

## Efficacy of HKCC against SARS-Cov-2 Omicron (BA.5) infection in hamsters

The efficacy of orally administered HKCC was assessed against SARS-CoV-2 Omicron (BA.5) infection in hamsters. Outbred Syrian hamsters were challenged intranasally with the SARS-CoV-2 Omicron (BA.5) variant (1.67 × 10⁵ PFU/

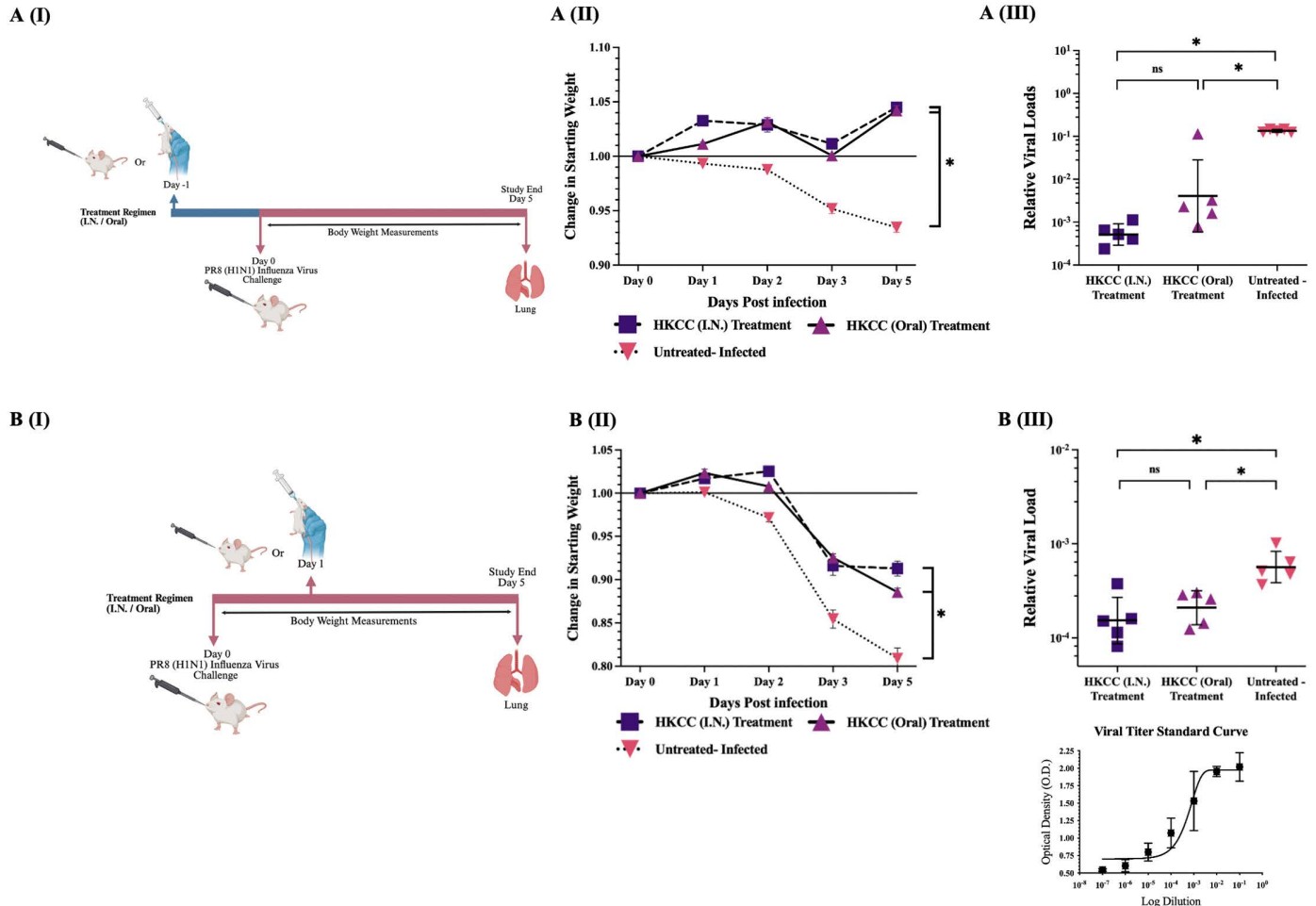

**Fig 4. Efficacy of HKCC against influenza (H1N1) infection in mice.** Female BALB/c mice (n = 5 per group) were treated with HKCC (50 × 10⁶ CFU/ mice) intranasally (i.n.) or orally. (**A**) Twenty-four hours before (Scheme created with Biorender, https://BioRender.com/az4bysi) or (**B**) twenty-four hours after (Scheme created with Biorender, https://BioRender.com/b5q1znm) intranasal challenge with the mouse-adapted PR8 (H1N1) influenza virus (60 PFU/mouse). (**I**) HKCC treatment regimen and sample collection timeline. (**II**) Weights of individual mice from the day of infection (day 0) to five days post-infection (days 0-5). (**III**) Viral loads in lung homogenates. The standard curve for H1N1 viral titers is shown (bottom right). Data are presented as mean ± standard error of the mean (SEM), with each data point representing an individual mouse. Statistical significance (*$p \leq 0.05$) was determined using one-way ANOVA followed by a Šidák post hoc test.

hamster) on day 0 and treated with HKCC (50 × 10⁶ CFU/hamster) three times on days -1, + 1, and +3, as described in Fig 5A. Five days after infection, viral titers (viral RNA) were measured in oral swabs (Fig 5B). Histopathological analysis was conducted to evaluate HKCC's ability to reduce or mitigate SARS-CoV-2 infections and protect against lung damage (Fig 5C and 5D). Notably, 3 out of 6 HKCC-treated hamsters showed complete clearance of the infection, with viral RNA levels approaching zero (Fig 5B) in the oral swabs. In contrast, the other three hamsters had viral RNA levels similar to those of untreated-infected hamsters (Fig 5B). Due to the variability in viral RNA levels, this group of outbred hamsters did not reach statistical significance between treated and untreated groups. However, lung histology analysis revealed that lung severity scores, as shown in Fig 5C, were significantly lower in all six hamsters treated orally with HKCC compared to untreated hamsters (Fig 5C and 5D). All HKCC-treated hamsters displayed milder lung pathologies characterized by

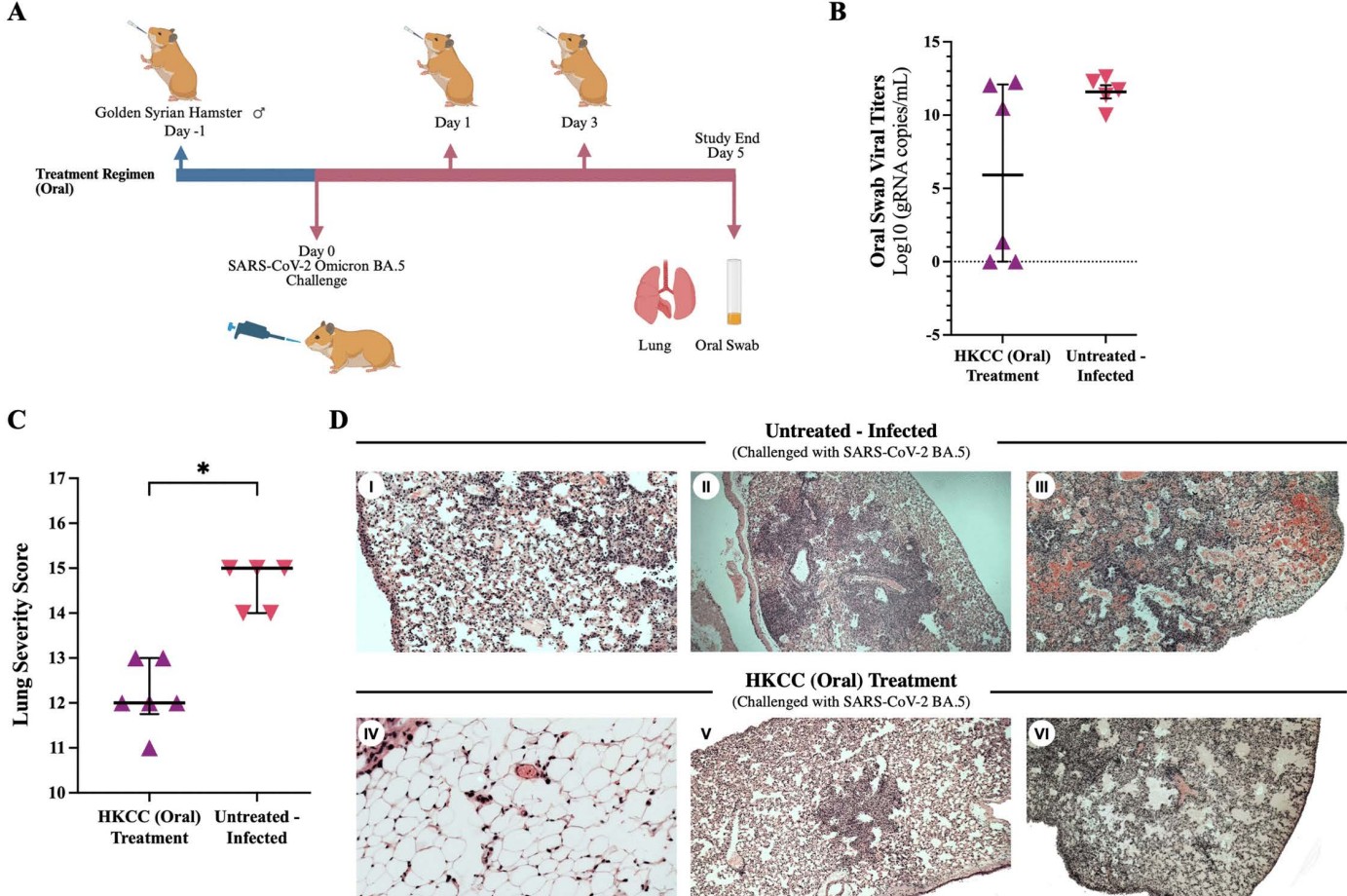

**Fig 5. Effectiveness of HKCC treatment against SARS-CoV-2 Omicron (BA.5) infection in hamsters. (A)** Schematic timeline of the HKCC treatment regimen and sample collection (Created with Biorender, https://BioRender.com/ua9vizl). Syrian hamsters were treated with HKCC ($50 \times 10^6$ CFU/hamster) orally on days -1, + 1, and +3, while the control group received no treatment (n = 5-6 per group). On day 0, hamsters were challenged intranasally with a $TCID_{50}$ dose ($1.67 \times 10^5$ PFU/hamster) of the SARS-CoV-2 Omicron (BA.5) variant. **(B)** Five days post-infection, SARS-CoV-2 genomic RNA copies were quantified in oral swabs. **(C)** Lung pathologies were assessed from H&E-stained lung sections collected five days post-infection, and severity scores were assigned using a standardized criterion for SARS-CoV-2–induced pathology in hamsters [26,82]. **(D)** Images of H&E-stained lung sections from untreated-infected **(I–III)** and HKCC-treated **(IV-VI)** groups. Data are presented as median ± interquartile range (IQR), with each point representing an individual hamster. Statistical significance (*$p \leq 0.05$) was determined using the Kruskal-Wallis test followed by Dunn's multiple comparisons test.

unobstructed airspaces, fewer patchy lesions spread throughout the lungs, and hemorrhages (Fig 5D IV-5D VI), compared to untreated, infected hamsters (Fig 5D I-5D III).

## Discussion

For respiratory infections worldwide, it is crucial to explore innovative host-immune-based therapies that can manage and/or reduce initial or ongoing infections, prevent or lessen disease severity, and improve the effectiveness of potential drug treatments [18]. Although the exact role and detailed regulation of various immune mechanisms that protect against pulmonary pathogens are not fully understood, it has become clear that both innate and adaptive immunity play important roles in mounting a successful immune response [29]. Especially in respiratory infections, the first line of immune defense

provided by innate components is vital for the host [22]. Therefore, an unconventional host-directed approach could be guided by relatively unexplored innate immune mechanisms that may be linked to host protection.

Previously, other beneficial microbes have been studied for their potential protective effects against respiratory infections. For example, the protective effect of postbiotic antimycobacterial metabolites derived from Lactic acid bacteria fed to wild boars was highly effective in reducing TB transmission [30]. Likewise, intranasal delivery of *Lactobacillus murinus* stimulated local pulmonary T-cell functions and offered protection [31]. Additionally, *L. casei,* when combined with anti-TB treatment, significantly reduced plasma levels of inflammatory cytokines in TB patients [32]. Interestingly, oral bacteria from a healthy human, when introduced into mice via the intranasal route, trigger protection against influenza and SARS-CoV-2 by stimulating antibody responses [33]. In this study, we aimed to investigate and leverage the contribution of innate immune responses to disease control across three infection models using a novel immune modulator, HKCC.

Innate lymphoid cells (ILCs), the innate counterparts of T cells, are relatively scarce compared to adaptive lymphocytes in lymphoid tissues; however, they are enriched at barrier surfaces such as skin, lungs, and intestines, as well as in adipose tissue and some mucosal-associated lymphoid tissues [18,34]. ILCs are strategically located at mucosal surfaces to interact with commensal microbiota and invading pathogens, helping to establish immune homeostasis and/or initiate immediate protective responses. They are functionally positioned upstream of adaptive immune responses [18]. Consequently, ILCs are key regulators of early immune responses and influence the subsequent development of protective responses as the disease progresses. We observed that mucosal administration of HKCC significantly reduced bacterial burden in the lungs, liver, and spleen, as well as lung pathology severity, in *Mav*-infected mice (Fig 1). To evaluate the effectiveness of immediate and trained innate immunity against *Mycobacterium* infection, two temporal cohorts were studied. We hypothesized that immediate prophylactic treatment would trigger rapid innate effector mechanisms, while a longer timeline would facilitate the development of trained innate immunity through functional, metabolic, and/or epigenetic reprogramming of myeloid cells. Exposure to microbial products about one week before the challenge has been shown to induce trained immune responses [35,36]. Therefore, we extended the treatment window to -21 and -8 days to further characterize trained immune components in response to mycobacterial infections. Among the two intranasal dose schedules, the shorter schedule (days -8 and -1) seemed to produce a better effect. These findings suggest that the immunomodulatory functions of HKCC mainly target rapidly acting innate immune mechanisms. We also examined ILCs in both treated and untreated groups to assess HKCC's effect on ILCs. An increased number of ILCs and NK cells was observed in the intranasal HKCC treatment groups (Fig 2A), which may partially contribute to the reduced bacterial loads across various organs and decreased lung pathology. Previous research suggested a protective role for ILCs in *Mtb* infection, as seen in a study involving 44 subjects with active TB—including both drug-sensitive and drug-resistant cases—where a significant reduction in ILC populations was observed compared to healthy controls [37]. Therefore, intranasal HKCC administration appears to induce ILCs that could play a role in disease control.

To assess the functionality of ILCs in treated versus untreated groups, we analyzed the expression of transcription factors and cytokine profiles of ILCs. In the group that received HKCC intranasally 8 and 1 days before infection, there were increased levels of pro-inflammatory cytokines and master regulator transcription factors of ILCs, RORγt and T-bet (Fig 2A III and 2A IV). Interestingly, this group showed the highest levels of protective cytokines, IL-17A and IL-13, and a moderate amount of IFNγ compared to the group treated intranasally with HKCC on days 21 and 8 before infection. However, both treatment groups showed a balanced cytokine profile compared with the untreated-infected control, which had fewer ILCs and a lower overall cytokine profile. The literature suggests a conflicting role of interferon in mycobacterial infection [38]. IFNγ production is not detrimental to the host, production of IFNγ during the early stages of respiratory infections is associated with protection against mycobacteria and acute viral respiratory infections [39,40]. Interferons I/II are required for recruiting NK cells into the lungs and activating them. It has been reported that mice with impaired interferon signaling pathways harbor significantly higher bacterial loads per cell and exhibit localized inflammatory lesions compared to control mice [41]. Similarly, IL-17A also displays a dual nature regarding the management of mycobacterial infection. A

recent study found that IL-17A production correlates with enhanced early protection in mice infected with the hypervirulent *Mtb* strain HN878 [42]. Our results also demonstrate that HKCC treatment involves transcriptional immune modulation that benefits the host, as shown by increased RORγt and T-bet expression following HKCC intranasal treatment on days 8 and 1 before the challenge. Although -21 and -8 days of treatment showed reduced expression of RORγt, IL-17A, and IL-13 compared with immediate treatment before infection, they resulted in the highest number of ILCs, suggesting sustained innate immune responses [43]. Therefore, administering the immunomodulatory agent HKCC via the nasal route enhances ILCs in the lungs and improves their functional properties through cytokine and transcription factor expression.

Mature ILCs exhibit significant phenotypic and functional heterogeneity and plasticity. Broadly, three groups of ILCs—ILC1, ILC2, and ILC3—have been classified based on surface markers, transcription factors, and effector cytokines [34]. *Ex vivo* restimulation of splenocytes from intranasally treated mice with HKCC or *Mav* lysate showed increased RORγt expression compared to those from untreated or orally treated mice (Fig 2B). These findings suggest that innate cells in the splenocytes of intranasally HKCC-treated groups gained the ability for enhanced or trained responses and activated the protective ILC2/3 signaling pathways. Upregulation of RORγt is known to be crucial for maintaining tissue homeostasis, mucosal integrity, pathogen defense, and regulation of inflammatory responses [44]. Additionally, one study indicated that RORγt stimulators could target ILC2 and ILC3, ultimately promoting strong mucosal defenses against pathogens [34].

Next, we hypothesized that HKCC treatment could influence the recruitment, migration, and activation of myeloid innate cells in the lungs. Existing literature has reported interesting patterns of CD11b$^+$ neutrophils and bacterial burden in specific models of mycobacterial infections [28]. The role of neutrophils in TB infections is complex and controversial; they can promote both disease progression and protection [45]. In our experiments, bacterial load analysis correlated with neutrophil (CD11b$^+$ Ly6G$^+$) infiltration in the lungs and spleens; however, the route of HKCC treatment impacted neutrophil infiltration in the lungs (Fig 2C). Intranasal administration of HKCC on days 8 and 1 before challenge substantially reduced the number of CD11b$^+$Ly6G$^+$ cells migrating to the lungs. This group showed the lowest bacterial load and lung severity score. In the spleen, regardless of the HKCC administration route, all HKCC-treated groups had fewer resident CD11b$^+$Ly6G$^+$ granulocytes than the untreated-infected control. These results suggest that overall mucosal HKCC administration can modulate innate immune responses against disseminated infection. The CD11b$^+$Ly6G$^+$ cell population has been identified as the primary niche for bacteria in the spleen [46] and lung alveoli during *M. bovis* infection [47]. Neutrophils are predominantly present in the lungs of non-human primates with TB infection [48]. Additionally, literature supports the idea that the accumulation of CD11b$^+$Ly6G$^+$ granulocytes in the lungs correlates positively with bacterial burden, and depleting these cells reduces bacterial load in the H37Rv-infected mouse model [28]. Even anti-Ly6G$^+$ treatment partly restored T-cell functions and improved lung pathology and mouse survivability in TB infection [47]. Neutrophils in *Mtb*-infected lungs show altered mitochondrial metabolism, characterized by increased lipid intake, which promotes *Mtb* growth and creates a nutrient-rich environment [48]. It has been demonstrated that recombinant IL-22 treatment or adoptive transfer of ILC3 can suppress neutrophil accumulation near alveoli and reduce neutrophil elastase 2 production, thus modulating excess lung inflammation in T2D mice infected with *Mtb* [49]. It is plausible that mucosal HKCC treatment could regulate neutrophil trafficking via the ILC3 pathway, thereby controlling inflammation, pathogen persistence and growth, and ultimately protecting the host from chronic infection. The functional properties of these migrated cell populations need further investigation to understand their impact on *Mav* infection.

The infiltrated granulocytes can both promote or suppress mycobacterial growth depending on their functional attributes. For instance, Ly6G$^+$ cells are a potential source of IL-10 secretion, and it was found that IL-10 secretion is not significantly linked to bacterial burden [28]. In our study, we observed that the CD11b$^+$ subset in the untreated-infected group was highly associated with IL-10 expression in both lungs and spleen (Fig 2C and 2D), showing higher bacterial loads and more severe lung pathology. Notably, the groups that received HKCC 8 and 1 days before challenge via intranasal or oral routes exhibited remarkably lower IL-10 levels in the lungs and no IL-10 expression in the spleen. Although IL-10 was higher in the CD11b$^+$ subset of the intranasal HKCC treatment group on days 21 and 8 before challenge, this group had

significantly reduced IL-10 expression in CD11b+Ly6G+ cells in both lungs and spleen. Overall, HKCC treatments led to suppressed IL-10 production in CD11b+Ly6G+ cells across all groups. In both groups that received HKCC on 8 and 1 days before challenge via intranasal and oral routes, regardless of levels of IFNγ, IL-12, and TNF expression, reduced IL-10 levels were associated with the development of strong immune responses to combat the infection. These results highlight that immediate mucosal treatment with HKCC can induce more resilient innate immune responses than treatment at earlier times. The group that received HKCC on days 21 and 8 before challenge still showed reduced immunopathology and lower CFU counts in the organs compared to untreated infected controls, and trained innate immunity may be at least partially responsible. Similarly, treating with beta-glucan before various infections boosted immunity and provided broad-spectrum protection against infections, including *Mtb*, *Staphylococcus aureus*, *Listeria monocytogenes*, *Escherichia coli*, *Citrobacter rodentium*, and *Pseudomonas aeruginosa* [35,36]. Consistent with our study, they showed that intraperitoneal administration in animals significantly reduced pulmonary CFU at 28 days post-infection and notably increased survival around 1 year after the *Mtb* challenge [35]. This indicates that HKCC treatment at -21 and -8 days induces trained innate immunity in the host and provides long-term protection. However, this long-term immunity shows a dampened response compared to immediate treatment before infection. This pattern of response may reflect more beneficial trained innate immune responses over an extended period. The detrimental effect of IL-10 on the host's response to mycobacterial infection has been previously studied and is mostly reported as a suppressor of antimicrobial responses *in vitro* [50], in mice [51], and humans [52]. Accordingly, our results are consistent with these studies; increased IL-10 appears to be associated with higher bacterial burden during *Mav* infection, and modulation of IL-10 levels offers protection.

The importance of maintaining a regulated level of pro-inflammatory cytokines in controlling mycobacterial infection with less pathology was well described by Gupta and colleagues [53]. In this context, we examined the cytokine expression patterns of the treated and untreated groups in the CD11b+ and CD11b+Ly6G+ populations (Fig 2C and 2D). We observed that the untreated-infected group exhibited extremely high levels of IFNγ, IL-12, TNFα, and IL-10 expression in CD11b+ and CD11b+Ly6G+ populations in both lungs and spleens, which may impair the development of protective immunity. Conversely, HKCC treatment groups in both CD11b+ and CD11b+Ly6G+ populations showed modulation of these pro-inflammatory cytokines. Proinflammatory cytokines like IFNγ, IL-12, and TNFα play a functional role in mycobacterial control [41,46,54]. We observed significant changes in their production in both lung and spleen of HKCC-treated mice. With the onset of infection, myeloid cell-mediated IL-12 drives innate immune responses, including the induction of interferon responses, Th cell polarization, and the stimulation of NK cells and other T cells to secrete further IL-12 [54]. The beneficial role of TNFα is critical during both the early and chronic phases of disease [46,55], and it influences chemokine expression by CD11b+ cells, thereby affecting immune cell migration during mycobacterial infection [56]. The ability of HKCC to balance proinflammatory cytokines and suppress regulatory cytokines highlights its potential as a host-beneficial immunomodulatory agent.

Previously, Eberl and colleagues suggested that fine-tuning ILCs could be an excellent opportunity for both prophylactic and therapeutic treatment of diseases; moreover, they noted that ILCs help maintain balanced immune responses without overactivating or overwhelming the immune system [34]. In line with this, we observed that HKCC treatment administered on a shorter timeline, specifically days 8 and 1 before infection, led to distinct and moderate innate immune responses, including ILCs and myeloid cells, in a way that benefits the host. A moderate level of IFNγ and reduced levels of IL-10, along with other effector cytokines, provided greater protection in groups treated with HKCC 8 and 1 days before challenge, whether delivered intranasally or orally. Therefore, based on this broader yet moderate innate immune activation caused by HKCC, we demonstrated the importance of balancing innate immune responses during the acute phase to prevent immunopathologies and enhance protection.

The role of humoral immune responses in intracellular pathogens, such as *Mycobacterium* spp., was previously underestimated. Increasing evidence supports the idea that antibody-driven immune responses are crucial in combating intracellular pathogens; however, this remains a controversial topic [57]. Distinct functional antibody profiles

have been observed between latent and active TB patients, with stronger antibody-mediated protective responses in latent TB, including macrophage activation, NK cell-mediated antibody-dependent cell cytotoxicity (ADCC), and intracellular bacillus killing by macrophages [58]. Notably, the antibody titers observed in the early stages of infection enhance phagocytosis, direct killing through phagolysosomal fusion, and allow processed antigens from the pathogen to be presented, thereby initiating cellular immune responses [59]. Interestingly, the interaction between ILCs and B cells, along with other hematopoietic cells, is now being described [34,60,61]. ILCs can promote B cell proliferation, increased antibody production, and isotype switching even independently of T cell responses [60,62]. Therefore, ILCs may influence the outcome of infection, especially mucosal infections, through downstream adaptive immunity as well. Given the importance of ILCs in antibody production, we examined the impact of HKCC on humoral responses against *Mav* infection.

Notably, we observed that mice treated with HKCC had increased systemic IgG and mucosal IgA levels specific for *Mycobacterium* (both *Mav* and BCG lysate), while antibody levels against HKCC were not elevated (Fig 3). Furthermore, we showed that IgG antibody titers had a significant negative correlation with bacterial load in all three organs, and a trend was observed between IgA and bacterial burden. However, there was a significant negative correlation between mucosal IgA against *Mav* and bacterial loads in the lungs (Fig 3D, 3I). Indeed, mucosal IgA responses have previously been shown to protect against respiratory pathogens. In particular, the presence of effector IgA at pathogen entry sites helps prevent pathogen invasion through its effector functions [63]. Thus, IgA likely plays a protective role and contributes at least in part to overall immunity. Additionally, systemic pathogen-specific IgG responses mediate cytotoxicity against bacteria and bacteria-infected cells [64]. We hypothesized that elevated functional ILCs may directly influence antibody production in HKCC-treated groups and that antibody responses could play a vital role in battling intracellular pathogens such as *Mycobacterium* spp. For example, type 2 ILCs stimulate B cells and induce IgM production independently of T cells, within the first week of pathogen exposure via the respiratory tract [60]. Type 3 ILCs have also been shown to promote site-specific mucosal and systemic antibody responses through B cells, using T-cell-independent pathways [61]. Our results strongly suggest that mucosal administration of HKCC induces acute-phase systemic and mucosal antibody production, possibly providing antibody-mediated protection for the host.

For acute respiratory viral infections such as influenza and SARS-CoV-2, a delicate balance between pro-inflammatory and anti-inflammatory responses is crucial for resolving infections, viral pathogenesis, lung damage, and disease progression. Historically, the antiviral effects of an early type I/II IFN response against influenza and SARS-CoV-2 are protective, while late IFN responses can drive disease progression [65]. During the initial phase of infection, respiratory viruses focus on maximizing their ability to infect host cells and increase viral replication. At the same time, viruses employ evasion strategies to dampen the innate antiviral immune response by inhibiting PRRs activation, interfering with the IFN response (IRF3/7, NF-κB), and disrupting JAK/STAT pathways [66]. In SARS-CoV-2 acute viral infections, viable virus can be detected in the lungs of hamsters 1–5 days after infection [39]. Similarly, mouse Influenza infection models are widely used in preliminary studies, often showing clinical signs within 2–3 days post-infection [67]. Therefore, the window for effective immunotherapy is early and narrow [68], when the virus is attempting to enhance its replication and spread without triggering the immune system. To evaluate the impact of innate immune responses triggered by HKCC on viral load reduction, HKCC was administered either 1 day before and/or early during infection, and samples were collected at 5 days post-infection (Figs 4 and 5). This proved to be a highly effective and appropriate treatment regimen, with HKCC demonstrating strong efficacy against both respiratory viruses. During oral treatment, microaspirates of the immunotherapeutic agent can reach the lungs, affecting immune responses in the respiratory tract [69]. Additionally, in the case of oral therapeutics, the primary site of absorption is the small intestine, and since the gut and lungs share a common mucosal system, there are exchanges of chemokines, cytokines, microbial metabolites, hormones, and immune cells between the gut and respiratory tract via the bloodstream [69]. Intranasally-delivered immunotherapeutics can exert immediate immunomodulatory effects, inducing a local respiratory mucosal response followed by a systemic immune response. Overall,

our studies show that early intervention with HKCC could be effective in reducing acute respiratory viral infections, such as influenza and SARS-CoV-2, and in preventing and/or treating chronic mycobacterial infections.

Immunotherapeutics are host-directed treatments that modify the innate and/or adaptive immune system to elicit protective immune responses, thereby resolving infections and reducing immunopathologies. An inefficient innate system, characterized by dysfunctional DCs, NK/NKT, and ILC populations, has been associated with worsened disease and lethal pneumonia in mice infected with Influenza and SARS-CoV viruses [70]. During Influenza infections, IFN-producing NK cells effectively clear the virus and decrease inflammation when exposed to an early, low dose of the virus, preventing it from replicating and causing overt infection [71]. In both influenza and SARS-CoV-2 infections, higher levels of ILCs correlate with better clinical outcomes and reduced pathology [72,73]. Mechanistically, cytokines produced by ILCs are important for shaping the pulmonary immune environment and act as primary determinants of pathogenesis. The hallmark of effective disease management is the complex, bidirectional crosstalk between ILCs and the adaptive immune system, which enables a coordinated, host-specific response. Therefore, strategies focused on ILCs for prevention and treatment are a promising approach to reducing the impact of respiratory viral infections.

Pro-inflammatory cytokines, including IFN, IL-1β, IL-6, and IL-12, help to inhibit viral replication and promote neutrophil- and CD8$^+$ T cell-mediated killing responses. A balanced, early immune response is crucial for controlling viral replication and preventing the shift to excessive, pathogenic inflammation. Our findings indicate that HKCC treatment helps establish this early viral control, which in turn inhibits progression to the 'cytokine storm' phase and reduces inflammatory lung damage in SARS-CoV-2-infected hamsters (Fig 5d). Similarly, early induction of these cytokines plays a significant role in controlling and eliminating SARS-CoV-2 infections [74]. If innate immune responses fail to control viral replication early, infection-induced cytokine storms occur later in the infection [75]. We hypothesize that HKCC treatment, which effectively reduces viral load, will result in milder pathology and prevent animals from progressing to the severe late-stage of infection. The animals in the viral infection study were monitored according to established protocols, and none showed adverse effects. Overall body weight loss and survival rates are key indicators of respiratory viral disease severity and reflect vaccine effectiveness [67,68,76]. A significant reduction in weight loss, at least a 50% decrease in viral load, the survival of all animals through the end of the study, and/or improved lung histopathology highlight the potential protective effect of HKCC. Therefore, HKCC treatment does not cause adverse reactions in influenza and SARS-CoV-2 infection models. Prior research on HKCC demonstrated its capacity to activate the innate immune system, marked by the upregulation of costimulatory molecules CD80/86 on monocytes, facilitating DC interactions with NK/NKT cells via IFN and IL-12, and the rapid production of IL-12, TNFα, IL-6, and IL-1β cytokines in the lungs within five hours of treatment [25]. IL-17A has been shown to decrease pulmonary complications in COVID-19 patients [74]. Our previous studies on HKCC have demonstrated increased production of GM-CSF, which helps improve clinical outcomes by stimulating alveolar epithelial repair during influenza infections. Overall, HKCC can induce a functional and controlled innate immune response, which is essential for effectively controlling microbial infections and providing protective mucosal immunity. Our current studies further illustrate its ability to modulate ILCs and other myeloid cells. Overall, HKCC's capacity to stimulate various innate immune mechanisms relevant to respiratory infections may contribute to its significant efficacy across multiple infection models. Understandably, the innate system relies on conserved molecular structures, known as pathogen-associated molecular patterns (PAMPs), to identify pathogens and engage a set of pre-programmed immune responses tailored to whether the pathogen is intracellular or extracellular. This allows the same innate immune mechanisms to broadly target a wide range of viruses and bacteria within a similar pathogen class, possibly explaining why innate responses induced by HKCC are effective against Influenza, SARS-CoV-2, and mycobacteria.

While these findings are significant, several limitations need to be addressed in future studies. This study used varying treatment and infection schedules across the three infection models. These timelines were selected based on previous literature to optimize the immediate effects of innate immunity and to assess pathology and disease severity, including histology, bacterial and viral loads, and body weight. However, to enhance the effectiveness of HKCC immunotherapy in

acute viral infections, further studies on dose, frequency, and timing are necessary. Additionally, we used immunocompetent hosts for mycobacterial infection; understanding the immunomodulatory potential of HKCC in immunocompromised hosts would be relevant to clinical settings. Our studies showed that HKCC treatment via both oral and intranasal routes did not generate antibody responses against HKCC; however, the impact of pre-existing immunity needs to be examined to determine HKCC's clinical potential. Finally, it would be prudent to compare the effects of other known immunomodulators directly with those of HKCC.

In summary, treatment with HKCC activates the major innate components, including ILCs and NK cells, while generating a protective cytokine profile. Furthermore, it controls the infiltration of potentially harmful granulocytes into infection sites, possibly via the ILC pathway, and boosts protective systemic and mucosal antibody titers in a host-beneficial manner. With these well-balanced innate immune activation mechanisms, our studies support HKCC as a potential broad-spectrum immunomodulatory agent for both prophylactic and therapeutic use against the world's leading respiratory infections.

## Materials and methods

### Ethics statement

All mouse experiments received approval from the University of Alberta's Animal Care and Use Committee for Health Sciences and were conducted in accordance with the guidelines of the Canadian Council of Animal Care (AUP 00003746 and AUP 00000212, respectively). Studies with Golden Syrian hamsters were carried out at the National Research Council Canada (NRC), following Canadian Council on Animal Care guidelines. Protocols and procedures received approval from the NRC's Human Health Therapeutics Animal Care Committee (Protocol 2020.06).

**Heat-killed caulobacter crescentus.** *Caulobacter crescentus* (ATCC) was cultured in a sterile environment at room temperature (RT) using PYE medium. Optical Density (OD) was measured frequently, and *C. crescentus* in the log phase of growth was further processed. Known amounts of colony-forming units (CFU) were centrifuged and resuspended in sterile phosphate-buffered saline (PBS) to reach the desired concentrations. Working stocks were aseptically stored at 4°C, and the required CFU for any immunization were heat-killed at 80°C for 1 hour before use.

**Animal treatment.** Mouse experiments were conducted using 7–8 week-old female BALB/c mice for *Mav* and H1N1 infection models. All animals were purchased from Charles River Laboratories and housed in designated pathogen-free facilities at the respective experiment sites. The SARS-CoV-2 study was performed on 6–7 week-old male Golden Syrian hamsters in a certified level 3 biocontainment facility at the National Research Council Canada (NRC). Upon arrival, animals were monitored and given a week to acclimate before starting the experiments. Healthy animals were randomly assigned to treatment groups according to the study timeline. Intranasal administration of HKCC ($50 \times 10^6$ CFU/30 μL total volume: 15 μL in each nostril) was performed under isoflurane gas anesthesia according to standard protocols [25]. For oral treatment groups, HKCC was administered via oral gavage at a dose of $50 \times 10^6$ CFU/200 μL.

***Mycobacterium avium* challenge and analysis of disease severity; Colony forming unit assay and histopathology analysis.** According to the literature, $1 \times 10^5 – 10^6$ CFU is recommended for intravenous inoculation to evaluate protective immune responses [77]. In previous experiments, we found that a dose of $5 \times 10^5$ CFU produced more consistent infections across organs and was detectable by the end of the study [25]. Therefore, the same dose of mycobacteria was used in this study. $5 \times 10^5$ CFU/50 μL of logarithmically grown *Mav* was administered intravenously at the designated time points following the treatment timeline (Fig 1A). Subsequently, the chronic stage of infection begins, and the bacterial load typically remains stable after 4 weeks post-infection, providing a clearer understanding of treatment effects [40,77,78]. Additionally, in mouse models, sustained innate immune responses can persist for up to one month after infection [43]. Thus, after four weeks, the challenged mice were humanely euthanized using $CO_2$, and samples were collected for further analysis. Individual mouse lungs, liver, and spleen were separately harvested and homogenized in sterile PBS. Serial dilutions of homogenates ($10^{-1}$ to $10^{-3}$) were cultured on 7H11 Middlebrook (Sigma-Aldrich) agar plates

supplemented with Middlebrook ADC enrichment (BD) and incubated at 37°C for 2–3 weeks. Colonies were counted, and bacterial loads were determined. One lung lobe from each of two randomly selected mice in each challenged treatment group was processed for hematoxylin and eosin (H&E) staining. Carefully harvested lung lobes were immediately placed in 10% formalin at RT for 2–3 days, then transferred to 70% ethanol. Next, tissues were embedded in paraffin, sectioned, and mounted on glass slides. Dried slides were stained with H&E and imaged using a Zeiss AXIO Observer.A1 Inverted Fluorescence Microscope at 10X magnification. To assess disease severity, a scoring system based on previously described methods was utilized [46]. The scale of granuloma formation described earlier was used to identify granuloma lesions [79], which were counted in each lung section. Tissues were evaluated and scored based on three factors: free air space unavailability, epithelial hyperplasia, and immune cell infiltration (increasing severity scores 1–5). The overall lung severity score was calculated by summing the scores for these three factors and the number of granuloma lesions.

**Flow cytometry analysis of immune cells.** The isolated lungs and spleens from each group were pooled and processed to create a single-cell suspension, providing enough cells for analysis. Therefore, each flow cytometry data plot in this study represents at least five biological replicates per group. Then, $2 \times 10^6$ cells per 1 mL per well were seeded in a 24-well plate separately and stimulated with PMA (50 ng/mL) and ionomycin (500 ng/mL) overnight at 37 °C and 5% $CO_2$. Afterward, brefeldin A (BFA; 1.5 µg/mL) was added, and cells were further incubated for 4 hours at 37 °C and 5% $CO_2$. Subsequently, splenocytes and lung lymphocytes were stained with an innate lymphoid cell panel (CD3-NovaFluor Red 710, CD127-PE-Cyanine5, CD49b-APC-eFluor 780, IFNγ-eFluor 450, IL-17A-FITC, T-bet-PerCP-Cyanine5.5, IL-13-PE-Cyanine7, and RORγt-PE-eFluor 610) and a myeloid cell panel (CD11c-PE-CYN5.5, CD11b-AF488, CD80-BV605, CD86-SB436, IFNγ-BUV737, TNFα-PE-CYN7, IL-12-PE, LY-6G-BV786, and IL-10-APC), following a standard flow cytometry protocol for surface [80] and intracellular [81] staining. The samples were analyzed on the same day as staining using Cytek Aurora Spectral Flow Cytometry (Cytek Biosciences). FloJo v10.10 software (BD) was used for analysis. Mean fluorescence intensity (MFI) values of each marker were adjusted to the cell number of the respective cell population analyzed and represented as heatmaps created by Morpheus. Adjusted MFI values were normalized to untreated-infected or untreated-uninfected samples, depending on the analysis requirements.

**Antibody enzyme-linked immunosorbent assay and correlation analysis.** Mycobacteria and *C. crescentus* cultured in 7H9 and PYE media, respectively, were measured by OD, and a defined number of CFU were centrifuged (3000 rpm for 20 min) and resuspended in sterile PBS. Subsequently, a solution with bacteria was heat-killed at 80°C for 30 min, followed by sonication at 37°C for 1 hour. The lysate supernatant was separated after centrifugation at 3000 rpm for 20 min and stored at -80°C until further use. Immunoplates were coated with *Mav*, *M. bovis,* or *C. crescentus* lysate at $1 \times 10^6$ CFU/50 µL per well and incubated overnight at 4°C. Then the standard ELISA protocol was followed [26].

**Influenza infection model.** Animals were randomly assigned and treated with HKCC ($50 \times 10^6$ CFU/mouse) either intranasally or orally, 24 hours before or after the mouse-adapted PR8 (H1N1) challenge (intranasally, 60 PFU/mouse) (Fig 4A I and 4B I). Weights were monitored daily for up to 5 days post-infection, and lung tissues were collected to evaluate viral load.

**MDCK cell-based ELISA for detecting H1N1 viral titers.** MDCK cells (ATCC: CCL-34) were cultured in MDCK growth media (1x DMEM, 10% Fetal Bovine Serum (FBS), 1 mM Sodium Pyruvate, 2 mM L-Glutamine, and 1x Pen Strep Antibiotic). These cells were plated at $2 \times 10^4$ cells per well and incubated overnight at 37°C with 5% $CO_2$. The next day, plates were examined under a microscope to ensure 80–90% confluency. Lung tissues from mice infected with mouse-adapted PR8 (H1N1) were homogenized. The lung homogenates were diluted to $10^{-1}$ and $10^{-3}$ and added to the respective wells. The plates were then incubated at 37°C with 5% $CO_2$ for 36 hours. After incubation, BFA (working dilution 1:2000) was added to each well (1:2000), and the plates were incubated for an additional 2 hours. Cells were then fixed using 1.4% cold paraformaldehyde (PFA) solution and incubated for 2 hours. Following fixation, plates were washed three times with 1x PBS (150 µL each), then treated with 0.3% Saponin solution for 20 minutes at 37°C with 5% $CO_2$. A blocking solution containing 1% FBS and 0.3% Saponin was added to each well and incubated for 1 hour at RT. Then incubated

with a detection antibody (1:1000), anti-influenza A virus Nucleocapsid antibody (Abcam, UK), in 0.3% Saponin for 1 hour at RT. Followed by the addition of a secondary antibody (1:1000), anti-mouse Ig-HRP conjugated (Southern Biotech, USA), and incubated for 1 hour at RT. Subsequently, True Blue TMB solution was added for 1 hour at RT, allowing color development. The reaction was then stopped with 1 N HCl and read immediately using a DTX 880 Plate Reader (Beckman Coulter) at 450 nm absorbance. A standard curve was generated by preparing H1N1 stocks diluted from $10^{-1}$ to $10^{-7}$ to interpolate the relative viral titers.

**SARS-Cov-2 hamster infection model.** Animals were randomly assigned to treatment groups, and researchers were blinded to the experimental groups. Hamsters received HKCC ($50 \times 10^6$ CFU/hamster) orally on days -1, +1, and +3 (Fig 5A). On day 0, hamsters were challenged with the SARS-CoV-2 Omicron (hCoV-19/South Africa/CERI-KRISP-K040013/2022, lineage BA.5; BEI Resources NR-56798) variant at a 50% tissue culture infectious dose ($TCID_{50}$) of $1.67 \times 10^5$ per animal, administered intranasally under anesthesia (ketamine and xylazine injection: 90 mg/kg and 8 mg/kg). Five days after the challenge, animals were euthanized with $CO_2$, and oral swabs and lungs were collected to assess viral titers and perform histology, respectively.

**Real-time polymerase chain reaction (RT-PCR).** Genomic viral RNA analysis of day 5 oral swabs was performed as previously described [26]. Briefly, collected swabs were processed, and viral RNA was extracted using the Quick-viral RNA kit (Cat. #R1035; Zymo Research). Total viral genomic RNA was quantified with the RT-qPCR kit (Luna Universal One-Step RT-qPCR; Cat. E3005S; New England Biolabs). RNA quantification and analysis were carried out using an Applied Biosystems QuantStudio 3 (Thermo Fisher Scientific) and Design and Analysis Software DA2 version 2.6.0, respectively.

**Histology.** Samples of H&E were prepared as previously described [26]. Histology assessments were conducted using a standardized reporting criterion for pathologies caused by SARS-CoV-2 infections in hamsters [26,82]. The evaluation focused on pathologies highly characteristic of SARS-CoV-2-induced pneumonia, including bronchial and peribronchial lesions, patchy distribution throughout the lungs, diffuse alveolar damage, hyaline membranes, cellular debris in alveoli, intra-alveolar fibrin deposition, air spaces (restricted [+]; open [–]), hemorrhage, edema, vasculature damage or collapse, alveolar and interstitial pneumonia, intra-alveolar cells, perivascular lymphocytic cuffing, and hyperplasia of bronchial epithelial cells (BECs) and alveolar epithelial cells (AECs). Lung histology slides displaying the assessed pathologies were assigned a score of +1 each, contributing to the overall severity score.

**Statistical analysis.** All analyses of results were performed using GraphPad Prism Software 10.2.0 (GraphPad Software). One-way or two-way analysis of variance with adjusted multiple comparisons was used to evaluate statistical significance. For the CFU assay and antibody ELISA, five biological replicates were analyzed individually. Spearman's correlation analysis was conducted to assess the relationship between CFU and serum/mucosal antibody titers. In all analyses, $p \leq 0.05$ was considered statistically significant.

## Supporting information

**S1 Data. Raw and analyzed data used in Figs 1–5.**
(XLSX)

## Acknowledgments

We thank Dr. Gabrielle Siegers from flow cytometry core of the FOMD, U of Alberta, for support.

## Author contributions

**Conceptualization:** Shanika Werellagama, Rakesh Kumar, Babita Agrawal.

**Formal analysis:** Shanika Werellagama, Raj S Patel, Babita Agrawal.

**Funding acquisition:** Babita Agrawal.

**Investigation:** Shanika Werellagama, Raj S Patel, Nancy Gupta, Satish vedi, Diana Duque, Jegarubee Bavananthasivam, Anh Tran, Babita Agrawal.

**Methodology:** Shanika Werellagama, Raj S Patel, Nancy Gupta, Satish vedi, Diana Duque, Jegarubee Bavananthasivam, Anh Tran, Babita Agrawal.

**Project administration:** Babita Agrawal.

**Resources:** Rakesh Kumar, Anh Tran, Babita Agrawal.

**Supervision:** Anh Tran, Babita Agrawal.

**Validation:** Rakesh Kumar, Anh Tran, Babita Agrawal.

**Visualization:** Shanika Werellagama, Raj S Patel, Babita Agrawal.

**Writing – original draft:** Shanika Werellagama, Raj S Patel, Babita Agrawal.

**Writing – review & editing:** Shanika Werellagama, Rakesh Kumar, Babita Agrawal.

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
