## [Decision Letter · Decision Letter 0]

30 Dec 2025

PLOS Pathogens

Dear Dr. Agrawal,

Thank you for submitting your manuscript to PLOS Pathogens. After careful consideration, we feel that it has merit but does not fully meet PLOS Pathogens's publication criteria as it currently stands. Therefore, we invite you to submit a revised version of the manuscript that addresses the points raised during the review process.

We look forward to receiving your revised manuscript.

Kind regards,

Anne Jamet

Section Editor

PLOS Pathogens

Anne Jamet

Section Editor

PLOS Pathogens

Editor-in-Chief

PLOS Pathogens

orcid.org/0000-0003-2946-9497

Editor-in-Chief

PLOS Pathogens

orcid.org/0000-0002-7699-2064

**Journal Requirements:**

At this stage, the following Authors/Authors require contributions: Babita Agrawal, Shanika Werellagama, Raj S Patel, Nancy Gupta, Satish vedi, Diana Duque, Jegarubee Bavananthasivam, Rakesh Kumar, and Anh Tran. Please ensure that the full contributions of each author are acknowledged in the "Add/Edit/Remove Authors" section of our submission form.

Potential Copyright Issues:

i) Figures 1A, 4A, 4B, and 5A. Please confirm whether you drew the images / clip-art within the figure panels by hand. If you did not draw the images, please provide (a) a link to the source of the images or icons and their license / terms of use; or (b) written permission from the copyright holder to publish the images or icons under our CC BY 4.0 license. Alternatively, you may replace the images with open source alternatives. See these open source resources you may use to replace images / clip-art:

5) In the online submission form, you indicated that The data supporting the findings in this article will be made available by the authors, without undue reservation, upon reasonable request.. All PLOS journals now require all data underlying the findings described in their manuscript to be freely available to other researchers, either

1. In a public repository

2. Within the manuscript itself

3. Uploaded as supplementary information.

2) If any authors received a salary from any of your funders, please state which authors and which funders..

7) Please send a completed 'Competing Interests' statement, including any COIs declared by your co-authors. If you have no competing interests to declare, please state "The authors have declared that no competing interests exist". Otherwise please declare all competing interests beginning with the statement "I have read the journal's policy and the authors of this manuscript have the following competing interests"

**Reviewers' Comments:**

Reviewer's Responses to Questions

**Part I - Summary**

Reviewer #1: In the current study by Werellagama et al., the authors have demonstrated the potential of heat killed Caulobacter crescentus (HKCC), a freshwater, non-pathogenic bacterium as an innate immunomodulatory agent. They showed that the HKCC treatment in respriratory infection animal models can induce robust localized and systemic mucosal immunity through comprehensive set of immunological and histopathological assays and imaging. They have shown that the HKCC treatment also bolsters adaptive immunity especially in viral infections like flu and SARS-CoV2. In summary, their data suggest both preventive and to some degree post-infection control of respiratory pathogens more effectively by boosting innate and adaptive immune responses. The study is clinically relevant and demonstrate the potential of immunomodulatory agents to boost mucosal immunity for better infection control.

Reviewer #2: Werellagama et al investigated a host-directed immunotherapeutic strategy against respiratory bacterial (Mycobacterium avium, Mav) and viral (SARS-CoV-2 and influenza) infections in animal models. This strategy used heat-killed Caulobacter crescentus (HKCC) as an innate immune modulator. They found that intranasal administration with HKCC stimulates innate lymphoid cells and neutrophils while enhances antigen-specific antibody production, resulting in reductions in both bacterial and viral loads and significantly milder disease progression. The authors indicated that HKCC may offer an effective, ready-to-use strategy to enhance the host defence against a broad range of bacterial and viral respiratory pathogens, warranting future clinical trials. There are, however, some major issues to be carefully addressed.

**Part II – Major Issues: Key Experiments Required for Acceptance**

Reviewer #1: (No Response)

Reviewer #2: First, it is unclear if HKCC can be generalised for clinical development against broad respiratory bacterial and viral infections. The data in figure one is unclear on how these time points were selected for intranasal administration. Since animals that received HKCC intranasally 21 and 8 days before challenge, along with untreated-infected controls, showed a more inflammatory profile with increased type 1 ILCs than ones received HKCC intranasally 8 and 1 days before challenge, this finding indicates a complex situation for clinical use of such innate immune modulator. The timing, dose and frequency for real world use would be hard to determine. It is, therefore, necessary to solve the underlying mechanism for a consistent innate immune modulatory effect.

Second, the authors did not address a critical issue on the impact of host pre-existing immunity. Since HKCC is common, the authors may need to determine if host pre-existing immunity to HKCC would have a major influence on the efficacy of such an intervention. To this end, if the HKCC intranasal administration induces anti-HKCC immunity, it is necessary to determine if the induced immunity would prevent the future efficacy of the same product.

Third, during the experiments, the authors used three infection models. They, however, did not provide convincing data if these models engaged the same mechanism of protection. This is a critical issue because it is known that some people may generate worse cytokine storms than others after SARS-CoV-2 infection. If HKCC generates worse inflammatory responses in these individuals, the safety evaluation should be done in more relevant clinical model.

**Part III – Minor Issues: Editorial and Data Presentation Modifications**

Reviewer #1: Rationale for bacterial choice and novelty:

The rationale for selecting Caulobacter crescentus over other bacterial genera is not sufficiently justified beyond its non-pathogenic nature. In principle, bacteria that more closely mimic respiratory pathogens might have been more appropriate for inducing relevant mucosal immune responses. Moreover, while the concept is interesting, it is not entirely novel. Previous studies have demonstrated that bacteria or bacterial extracts (e.g., OM-85, Lactobacillus species) can induce mucosal immunity in the respiratory tract and gut. These studies should be incorporated into the Introduction or Background, with a clearer discussion of how the current work advances or fills gaps in existing knowledge. In addition, the study lacks comparative controls using bacteria from other genera or established immunomodulatory agents. Including one or two well-characterized adjuvants would have strengthened the study design and contextualized the observed effects of HKCC.

Relevance to immunocompromised hosts:

Although HKCC treatment markedly improves disease outcomes in the Mycobacterium avium infection model, it remains unclear whether such robust immune induction would occur in immunocompromised settings, where mycobacterial infections are most clinically relevant. This limitation should be acknowledged, and the potential variability of HKCC efficacy in immunocompromised hosts discussed.

Durability and limitations of preventive and therapeutic efficacy:

The authors propose HKCC as both a preventive and adjunct therapeutic approach for respiratory infections. A major concern for the preventive strategy is the durability of the induced innate immune responses. The data clearly show that HKCC administration 24 hours prior to infection (8–1 day scheme) yields superior protection compared to the longer interval (21–8 day scheme), suggesting that the immune response may be relatively short-lived. This could represent a significant hurdle for preventive applications. Furthermore, the post-infection treatment approach appears less effective, as evidenced by comparatively modest disease control (e.g., body weight changes and viral titers in treated groups). These limitations should be explicitly discussed, as they warrant further investigation and optimization.

Variability in experimental design across infection models:

The treatment and infection schedules differ substantially across the three infection models used in the study. A clearer rationale for these variable experimental designs is needed, along with an explanation of how these differences impact data interpretation and cross-model comparisons.

Reviewer #2: Fig 1, is the protective efficacy dose-dependent? Why was only 5x105 PFU used for this experiment? The scale bar for each H.E. image is missing.

Fig 2, was there any statistical significance for T-Bet or other parameters measured?

Fig 3, did the lung IgA show any protective potency as compared with IgG?

Fig 4 & 5, did innate lymphoid cells and neutrophils correlate with protection?

PLOS authors have the option to publish the peer review history of their article (what does this mean? ). If published, this will include your full peer review and any attached files.

**Do you want your identity to be public for this peer review?** For information about this choice, including consent withdrawal, please see our Privacy Policy .

Reviewer #1: **Yes:** Swati Jain

Reviewer #2: No

**Figure resubmission:**

**Reproducibility:**



---

## [Decision Letter · Decision Letter 1]

12 Feb 2026

Dear Dr. Agrawal,

We are pleased to inform you that your manuscript 'Host-directed Broad-spectrum Immunotherapeutic Strategy for Respiratory Infections: Heat-killed Caulobacter crescentus (HKCC) as an Innate-immune based Biotherapeutic/Postbiotic' has been provisionally accepted for publication in PLOS Pathogens.

Best regards,

Anne Jamet

Section Editor

PLOS Pathogens

Sumita Bhaduri-McIntosh

Editor-in-Chief

PLOS Pathogens

orcid.org/0000-0003-2946-9497

Michael Malim

Editor-in-Chief

PLOS Pathogens

orcid.org/0000-0002-7699-2064

Reviewer Comments (if any, and for reference):

Reviewer's Responses to Questions

**Part I - Summary**

Reviewer #1: The authors have thoroughly addressed the reviewers’ comments and incorporated the suggested revisions. Importantly, they have also added a clear discussion of the study’s limitations, which strengthens the overall rigor and transparency of the manuscript.

Reviewer #2: This study report a strategy using heat-killed Caulobacter crescentus (HKCC) as a potential innate immune modulator.

PLOS authors have the option to publish the peer review history of their article (what does this mean? ). If published, this will include your full peer review and any attached files.

**Do you want your identity to be public for this peer review?** For information about this choice, including consent withdrawal, please see our Privacy Policy .

Reviewer #1: **Yes:** Swati Jain

Reviewer #2: No

**Part II – Major Issues: Key Experiments Required for Acceptance**

Reviewer #1: (No Response)

**Part III – Minor Issues: Editorial and Data Presentation Modifications**

Reviewer #1: (No Response)

---

## [Editor Report · Acceptance letter]

Dear Dr. Agrawal,

We are delighted to inform you that your manuscript, "Host-directed Broad-spectrum Immunotherapeutic Strategy for Respiratory Infections: Heat-killed Caulobacter crescentus (HKCC) as an Innate-immune based Biotherapeutic/Postbiotic," has been formally accepted for publication in PLOS Pathogens.

Best regards,

Sumita Bhaduri-McIntosh

Editor-in-Chief

PLOS Pathogens

orcid.org/0000-0003-2946-9497

Michael Malim

Editor-in-Chief

PLOS Pathogens

orcid.org/0000-0002-7699-2064